# Computing the Thermal Efficiency of Autoclaves during Steaming of Frozen Prisms for Veneer Production at Changing Operational Conditions

Nencho Deliiski [1,*], Peter Niemz [2,3], Dimitar Angelski [1], Pavlin Vitchev [1] and Natalia Tumbarkova [1]

[1] Faculty of Forest Industry, University of Forestry, Kliment Ohridski Blvd., 10, 1797 Sofia, Bulgaria
[2] Institute for Building Materials, ETH Zürich, CH 8093 Zürich, Switzerland
[3] Department of Wood Science and Engineering, Luleå University of Technology, 931 77 Skellefteå, Sweden
[*] Correspondence: deliiski@netbg.com

**Abstract:** A methodology for the computation of the thermal energy efficiency of modes for the heat treatment of frozen wooden prisms in an autoclave with saturated water vapor at changing operational conditions has been proposed. The methodology includes computer simulations with two own-coupled unsteady models: one to calculate the 2D temperature distribution in the cross-section of prismatic wood materials during their heat treatment, and the second to determine the heat balance of industrial autoclaves for such wood treatment. Simulations were carried out in order to determine the duration, energy consumption, and thermal efficiency of different modes, caused by changed operational conditions, for the autoclave steaming of frozen beech prisms with industrial parameters in the absence and presence of dispatcher intervention. The influence of nine combinations between the time of dispatcher intervention and the degree of reduction of the constant maximum temperature from the 130 °C of the basic mode on the thermal efficiency of the autoclave was investigated. The results show that all studied dispatching interventions cause an increase in both the duration and the thermal efficiency of the modes. This efficiency in the modes at changing operational conditions has values between 68.7% and 74.6%, while the efficiency in the basic steaming mode is equal to 68.0%.

**Keywords:** frozen wooden prisms; autoclave steaming; changing operational conditions; dispatching intervention; energy consumption; energy efficiency; veneer production

## 1. Introduction

It is well known that for the decorative coating of details intended for the production of high-quality furniture as well as for the furnishing of office premises, the most frequently utilized practice is to use a veneer, obtained from variable wooden species, with different thickness, texture and color than the wood being heat treated [1–4]. In the manufacture of veneer, wood materials are subjected to heat treatment with saturated steam or hot water in various facilities in order to plasticize them before cutting the veneer [5–16]. This is determined by the circumstance that the heated wood has an increased deformation capability and is susceptible to cutting.

To reduce the duration and energy consumption of heat treatment, instead of using equipment operating at atmospheric pressure, autoclaves are used [11–14,16–27].

Heat treatment modes are usually presented in the literature sources, which are applied in practice to ensure the minimum possible duration of the wood heating process and maximum productivity of the equipment [5–14,28–36]. These sources completely lack data on modes in which dispatcher intervention increases their duration in order to ensure the necessary wood plasticity is achieved later than the usual time due to changed operational conditions. Only in [37,38] are results given for a computer-simulated study of the duration of modes for the steaming of non-frozen beech prisms in an autoclave and of the energy needed for heating the wood itself, when applied to the operations of various

dispatcher interventions. It was established that during the steaming of prismatic materials with a cross-section of $0.4 \times 0.4$ m, $u = 0.6$ kg·kg$^{-1}$ and $t_{w0} = 0$ °C in a basic mode with $t_{m\text{-}max} = 130$ °C = const and a duration of 9.15 h, maximum productivity of the autoclave is ensured at an energy consumption of 65.35 kWh·m$^{-3}$, which is required only for warming up of the wood. The dispatcher's lowering of $t_m$ from 130 °C to 100 °C in the 3rd, 5th and 7th h of the basic mode causes an increase in its duration to 14.15, 13.15 and 12.15 h, respectively, while the indicated energy consumption is reduced in all studied cases to about 58.0 kWh·m$^{-3}$. Information only on the duration of analogous modes with dispatcher intervention, intended for the heat treatment of frozen beech prisms with saturated water vapor, is published in [39]. The partial results are reflected below in the last column of Table 1.

In [14], it was found that during the steaming of beech materials with cross-sectional dimensions of $100 \times 100$ mm and $u = 0.6$ kg·kg$^{-1}$ in an autoclave for 7 h on separate modes with $t_m = 100$, 120 and 140 °C, the consumption of heat energy changes from 97.8 to 155.4 kWh·m$^{-3}$ at initial wood temperature $t_{w0} = 0$ °C; and from 135.1 to 196.3 kWh·m$^{-3}$ at $t_{w0} = -20$ °C. References [9,10] note that the efficiency of the heat treatment of wood materials with saturated water vapor in pits during veneer production does not typically exceed 18%.

Studying the impact of time- and size-varying dispatcher interventions on the duration and energy parameters of wood heat treatment modes has significant scientific and practical interests. In the event of organizational or technical problems in the production lines for the cutting and drying of veneer, it is required to extend the modes for the heat treatment of the wood until the problem is solved, while at the same time ensuring the necessary plasticity of the material. In such cases, it is necessary for the dispatcher or automated control system to intervene in a timely manner and to change the temperature-time parameters of the current mode in an appropriate way.

The proposal and application of a methodology for studying the influence of different dispatcher interventions in wood heat treatment modes will allow, in the future, for the development of software for advanced model-based systems for the automatic control of various types of such treatments [21,25,40–43]. Solving a task with such a great complexity and multifactorial nature [16,39] can only be done using adequate multiparameter temperature-time-energy models.

This paper presents a methodology for the application of two personal coupled non-stationary models for computing the thermal energy efficiency of different modes of autoclave steaming on frozen wood prisms intended for the production of veneer for the cases of time- and magnitude-variable dispatch interventions.

## 2. Materials and Methods

### 2.1. Materials for Research

Two coupled models, created and verified in [20], and the methodology suggested in [44] were used for simulated research of the thermal efficiency of modes for the steaming of frozen wooden prisms in an autoclave in cases of dispatcher intervention in the temperature-time modes' parameters.

The study was carried out with ice-containing beech (*Fagus sylvatica* L.) prisms above the hygroscopic range, which are commonly used in veneer production.

During numerical simulations with the mathematical models presented below, the following parameters of the prisms, which influence $\tau_{mode}$ and $\eta$, were set: $d \times b \times l = 0.4 \times 0.4 \times 2.5$ m, $t_{w0} = -20$ °C, $\rho_b = 560$ kg·m$^{-3}$, $u = 0.6$ kg·kg$^{-1}$, and $u_{fsp}^{293.15} = 0.31$ kg·kg$^{-1}$. Beech prisms of such dimensions and with a moisture content above the hygroscopic range are relatively often subjected to steaming in veneer production in practice. Such prisms, with an initial temperature of $-20$ °C, contain significant amounts of both free and bound water in a frozen state, the thawing of which will favor the increase of the differences in the durations between the individual steaming modes studied.

The simulations were performed under the following industrial parameters of a well-insulated wood steaming autoclave used in practice: $D = 2.4$ m, $L = 9.0$ m, $\gamma = 50\%$, and $q_{\text{source}} = 500$ kW. Detailed structural and thermo-physical characteristics of such autoclaves, as well as their application in the woodworking industry, is described in [12,14,20,22,23].

### 2.2. Modelling of the 2D Unsteady Temperature Change in Prisms

The following 2D model of the temperature change in prismatic wood materials during their heat treatment with saturated water vapor and subsequent cooling (conditioning) in an air medium before veneer cutting was created and verified in [20]:

$$c_{\text{w-eff1,2,3}} \cdot \rho_{\text{w}} \frac{\partial T}{\partial \tau} = \text{div}(\lambda_{\text{w-cr}} \text{grad } T) \tag{1}$$

$$\text{at } T(x, y, 0) = T_{\text{w0}} \tag{2}$$

and at the following prism surface temperatures:

■  during the steaming process:

$$T(x, 0, \tau) = T(0, y, \tau) = T_{\text{m}}(\tau) \tag{3}$$

■  during the subsequent conditioning process:

$$\frac{\partial T(x, 0, \tau)}{\partial y} = -\frac{\alpha_{\text{w-cond}}(x, 0, \tau)}{\lambda_{\text{w-cr}}(x, 0, \tau)}[T(x, 0, \tau) - T_{\text{air-cond}}(\tau)] \tag{4}$$

$$\frac{\partial T(0, y, \tau)}{\partial x} = -\frac{\alpha_{\text{w-cond}}(0, y, \tau)}{\lambda_{\text{w-cr}}(0, y, \tau)}[T(0, y, \tau) - T_{\text{air-cond}}(\tau)] \tag{5}$$

The defrosting process of ice-containing wood materials during steaming for the purpose of plasticization in the production of veneer takes place in three stages [14,20,32]. Figure 1 shows these three stages, as well as the symbols of the thermo-physical characteristics of the wood in each of them, for which it is necessary to have a mathematical description when solving the model (1)–(5).

The effective specific heat capacities of the wooden prisms during their defrosting, $c_{\text{w-eff1,2,3}}$, which participate in Equation (1), are described mathematically as follows [6,20,26,32,45]:

$$\text{First stage}: c_{\text{w-eff1}} = c_{\text{w-fr}} + c_{\text{ice-fw}} \tag{6}$$

$$\text{Second stage}: c_{\text{w-eff2}} = c_{\text{w-nfr}} + c_{\text{ice-fw}} \tag{7}$$

$$\text{Third stage}: c_{\text{w-eff3}} = c_{\text{w-nfr}} \tag{8}$$

where

$$c_{\text{w-fr}} = K_{\text{c-fr}} \frac{526 + 2.95T + 0.0022T^2 + 2261u + 1276 \cdot u_{\text{fsp}}^{272.15}}{1 + u} \tag{9}$$

$$K_{\text{c-fr}} = 1.06 + 0.04u + \frac{0.00075(T - 272.15)}{u_{\text{fsp}}^{272.15}} \tag{10}$$

$$c_{\text{ice-bw}} = \left(69.344T + 119.183T \cdot \ln \frac{T}{273.15}\right) \cdot \left(u_{\text{fsp}}^{272.15} - 0.12\right) \cdot \frac{\exp[0.0567(T - 272.15)]}{1 + u} \tag{11}$$

$$c_{\text{w-nfr}} = \frac{1}{1 + u} \cdot \left(2862u + 2.95T + 5.49u \cdot T + 0.0036T^2 + 555\right) \tag{12}$$

$$c_{\text{ice-fw}} = 3.34 \times 10^5 \frac{u - u_{\text{fsp}}^{272.15}}{1 + u} \tag{13}$$

$$u_{\text{fsp}}^{272.15} = u_{\text{fsp}}^{293.15} + 0.021 \tag{14}$$

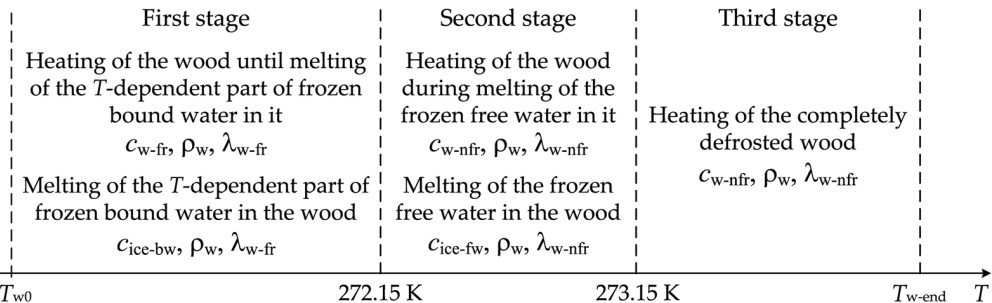

**Figure 1.** Stages of the wood defrosting process and thermo-physical characteristics of wood used in them.

The mathematical descriptions of the wood thermal conductivity, $\lambda_{w\text{-}cr}$, wood density, $\rho_w$, and heat transfer coefficient, $\alpha_{w\text{-}cond}$, given in [20,26,32,46,47] were used in solving models (1)–(5).

It can be noted that the experimentally established data for $\lambda_w = f(t,u)$ and $c_w = f(t,u)$ in the dissertations [6,48] were used as a basis in the mathematical descriptions of $\lambda_{w\text{-}cr}$, $c_{w\text{-}fr}$, and $c_{w\text{-}nfr}$, which were involved in Equations (1) and (4)–(12). These data are often used in both the European [8,10–14,28,47,49] and the American [29,50–56] literature sources, which consider approaches for calculating processes of the defrosting and/or heating of different wood materials.

### 2.3. Modelling of Thermal Efficiency of Modes for Steaming of Wooden Prisms in Autoclaves

To calculate this efficiency, it is necessary to have a mathematical description of the energy consumption of the entire autoclave and separately of the part of it that is used to heat the wood placed in the equipment.

In the simulations, the total energy consumed by the autoclave, $Q_a^n$, was calculated using the following unsteady model of its thermal balance, which was proposed and experimentally verified in [20]:

$$Q_a^n = Q_w^n + Q_{mb}^n + Q_{il}^n + Q_e^n + Q_{fv}^n + Q_{cw}^n \tag{15}$$

Dependence of all the components of the thermal balance on the multitude of influencing factors have been considered in [14,16,20].

The thermal energy consumption used for heating of the frozen prismatic wood materials at any time, $n \cdot \Delta\tau$, $Q_w^n$, was calculated by the following equation [20,32,57]:

$$Q_w^n = \frac{\rho_w}{3.6 \times 10^6 S_w} \cdot \left\{ \iint\limits_{S_w} \frac{c_{w\text{-}eff1,2,3}^n \, \text{at} \, T_{i,k}^n + c_{w\text{-}eff1}^n \, \text{at} \, T_{w0}}{2} \cdot \left( T_{i,j}^n - T_{w0} \right) \mathrm{d}S_w \right\} \tag{16}$$

where

$$S_w = \frac{d \cdot b}{4} \tag{17}$$

The thermal energy efficiency, η, of the separate steaming modes of prisms in an autoclave is equal to

$$\eta = 100 \frac{Q_{w\text{-}max}^n}{Q_{a\text{-}max}^n} \tag{18}$$

where $Q_{w\text{-}max}^n$ and $Q_{a\text{-}max}^n$ are calculated by the models as the maximum values of $Q_w$ and $Q_a$ for each of the autoclave steaming modes given in Table 1.

**Table 1.** Change in the temperature $t_{m1}$ and duration $\tau_{steam}$ of the studied autoclave steaming modes.

| Steaming Modes | Ist Stage of $t_{m1}$, °C | IInd Stage of $t_{m1}$, °C | Ist Stage of $t_{m1}$ $\tau_I$, h | IInd Stage of $t_{m1}$ $\tau_{II}$, h | $\tau_{steam}$ = $\tau_I + \tau_{II}$, h |
|---|---|---|---|---|---|
| Mode 0 | 130 | – | 13.9 | – | 13.9 |
| Mode 1 | 130 | 120 | 3.0 | 13.2 | 16.2 |
| Mode 2 | 130 | 120 | 7.0 | 8.7 | 15.7 |
| Mode 3 | 130 | 120 | 11.0 | 4.2 | 15.2 |
| Mode 4 | 130 | 110 | 3.0 | 15.7 | 18.7 |
| Mode 5 | 130 | 110 | 7.0 | 10.7 | 17.7 |
| Mode 6 | 130 | 110 | 11.0 | 5.7 | 16.7 |
| Mode 7 | 130 | 100 | 3.0 | 17.7 | 20.7 |
| Mode 8 | 130 | 100 | 7.0 | 12.1 | 19.1 |
| Mode 9 | 130 | 100 | 11.0 | 6.5 | 17.5 |

*2.4. Change in the Steaming Medium Temperature $T_m$ of Modes in Cases of Absence and Presence of Dispatcher Intervention*

In the numerical simulations with the models (1)–(5) and (15), the change of $T_m$ shown in Figure 2, whose mathematical description is presented in [14,20,58], is assumed.

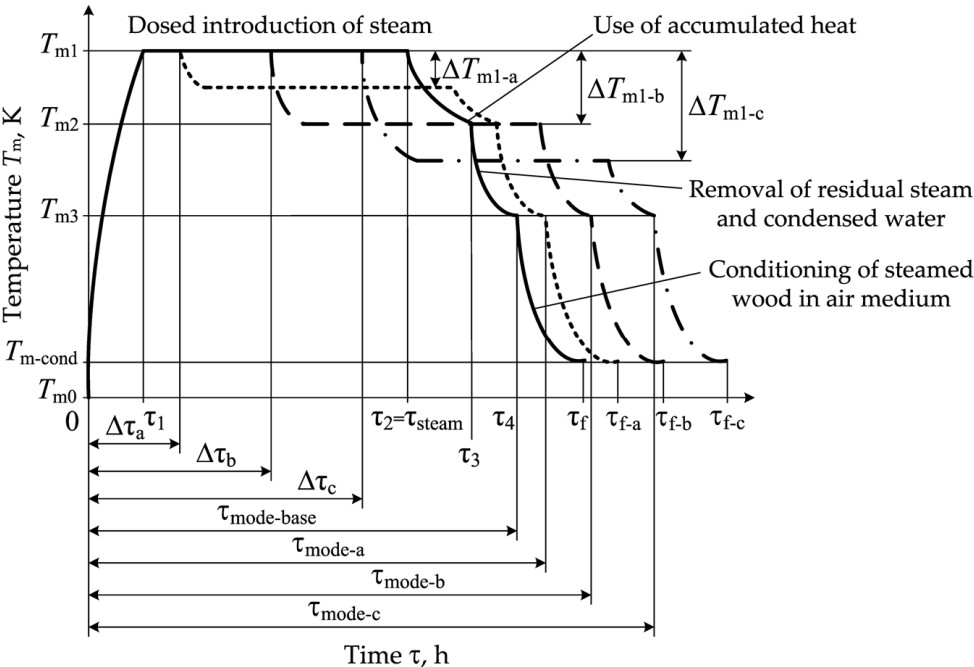

**Figure 2.** Change of $T_m$ during steaming modes in absence and presence of dispatcher intervention.

The following values of the parameters of the modes, whose symbols are given in Figure 2, were set: $T_{m0}$ = 253.15 K (i.e., $t_{m0}$ = −20 °C); $T_{m1}$ = 403.15 K (i.e., $t_{m1}$ = 130 °C); $\Delta T_{m1-a}$ = 10 K = 10 °C, $\Delta T_{m1-b}$ = 20 K = 20 °C, and $\Delta T_{m1-c}$ = 30 K = 30 °C; $T_{m2}$ = 383.15 K (i.e., $t_{m2}$ = 110 °C); $T_{m3}$ = 353.15 K (i.e., $t_{m3}$ = 80 °C); $T_{m-cond}$ = 293.15 K (i.e., $t_{m-cond}$ = 20 °C); $\Delta\tau_a$ = 3 h, $\Delta\tau_b$ = 7 h, and $\Delta\tau_c$ = 11 h. These values of the $T_m$ parameters fully correspond to those applied in the regimes for the industrial steaming of wood materials in the production of veneer [12,14,22].

With these values, one basic mode without dispatcher intervention (Mode 0) was calculated, as well as 9 modes with dispatcher intervention, which in Table 1 are described as Mode 1 to Mode 9.

Table 1 also provides the temperatures and durations of the two stages (before and after the dispatcher's intervention) of Mode 1 to Mode 9, and also the total duration of the introduction of steam into the autoclave, $\tau_{steam}$, for all of the investigated modes.

For joint numerical solving of the experimentally verified coupled models presented above and aimed at computation of the duration, thermal energy consumption, and energy efficiency of the modes given in Table 1, a personal software program was created in the Visual FORTRAN computing environment. Using this program, the modes shown in Table 1 were developed.

An explicit finite-difference scheme was used to transform the individual model equations into a FORTRAN-friendly programming form, which excludes any simplifications of the models.

The development of the modes consisted in selecting the values of the parameters shown in Figure 2 and Table 1, and that at the end of the conditioning of the steamed prisms, to ensure the required good plasticity of the wood before cutting the veneer. The degree of plasticity of the steamed prisms is assessed when the temperature distribution in their central cross-section falls completely within the optimal limits recommended for beech wood in veneer production $t_{opt-min}$ = 62 °C and $t_{opt-max}$ = 90 °C [14,59].

Simultaneously with the solution of the models, computations of $t_{w-avg}$, $Q_w$, and $Q_a$ were carried out. After determining the maximum values of $Q_w$ and $Q_a$, the energy efficiencies η of each of the studied modes were calculated.

## 3. Results

### 3.1. Computing the 2D Unsteady Temperature Change in Prisms during Studied Modes

In Figure 3, as an example, the calculated change in $t_s$, $t_{w-avg}$, and $t$ of 2 representative points $t_1$ (with coordinates $d/8$, $b/8$) and $t_2$ (with coordinates $d/2$, $b/2$) of the prisms during Mode 0, Mode 5, and Mode 8 is presented. Analogous figures for modes 1, 3, 4, 6, 7, and 9 can be seen in [39].

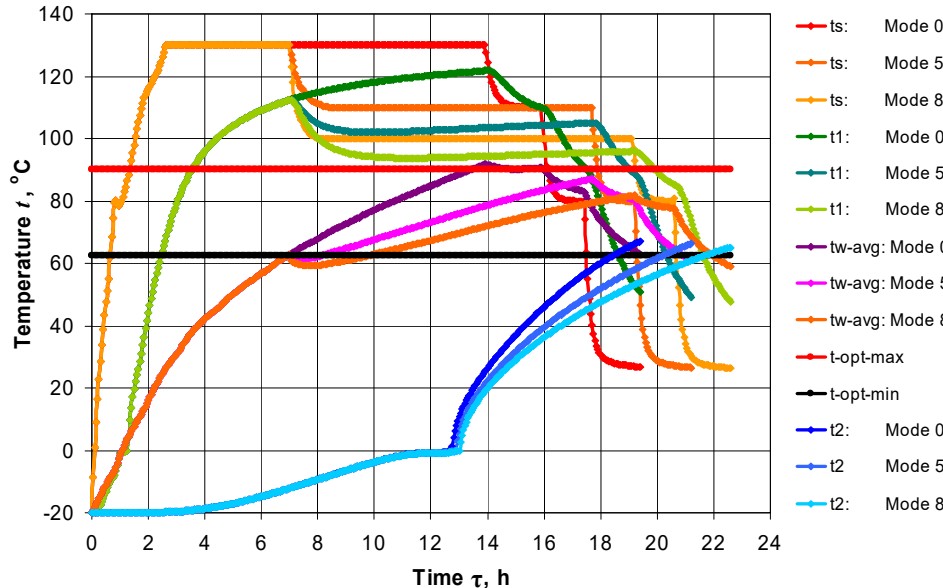

**Figure 3.** Changes in $t_s$, $t_{w-avg}$, $t_1$, and $t_2$ of the prisms during steaming Mode 0, Mode 5, and Mode 8, depending on τ.

### 3.2. Computing the $Q_a$, $Q_w$, and η for the Cases of Absence and Presence of Dispatcher Intervention in Steaming Modes

In Figures 4–6, the computed values of the energies $Q_a$ and $Q_w$ during the all studied modes is presented.

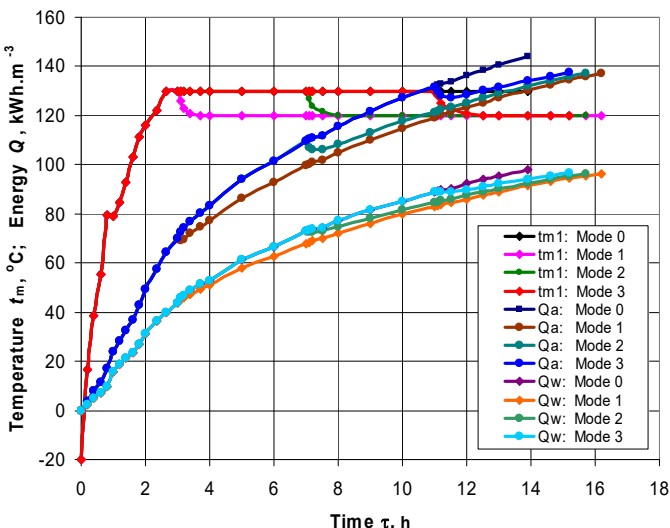

**Figure 4.** Changes in $t_m$, $Q_a$, and $Q_w$ during steaming modes 0, 1, 2, and 3, depending on $\tau$.

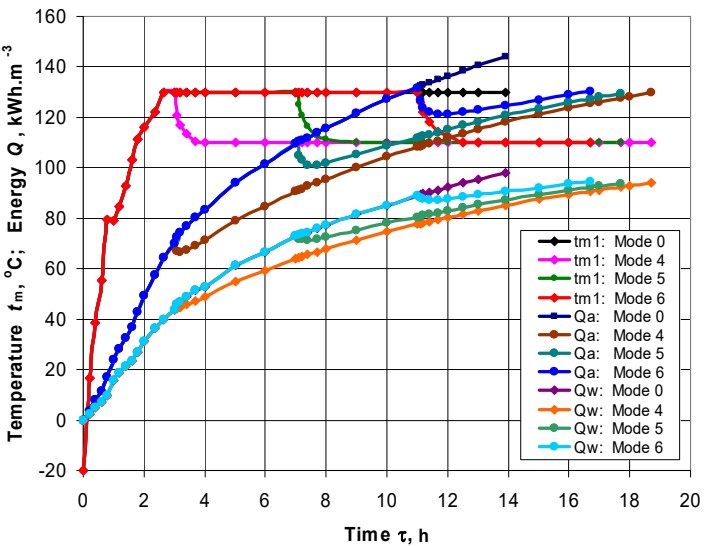

**Figure 5.** Changes in $t_m$, $Q_a$, and $Q_w$ during steaming modes 0, 4, 5, and 6, depending on $\tau$.

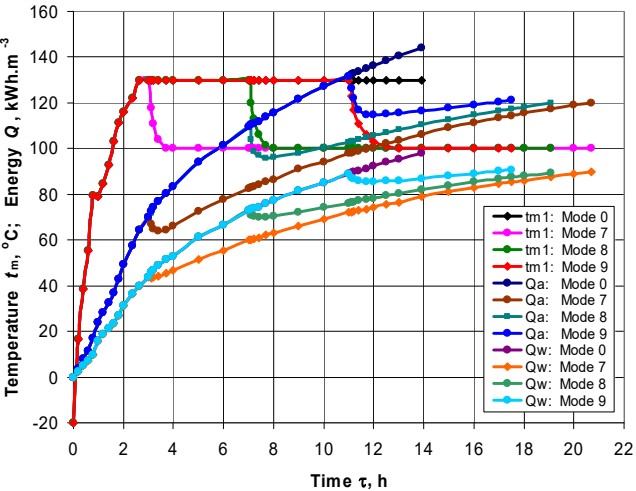

**Figure 6.** Changes in $t_m$, $Q_a$, and $Q_w$ during steaming modes 0, 7, 8, and 9, depending on $\tau$.

In Table 2, the change in $\tau_2 = \tau_{\text{steam}}$, and $\tau_4 = \tau_{\text{mode}}$ (see Figure 2) of all modes, and also of the $t_{\text{w-avg}}$, $Q_{\text{w-max}}$, $Q_{\text{a-max}}$, and $\eta$ is given.

Figure 7 shows the change in $Q_{\text{w-max}}$ and $Q_{\text{a-max}}$ for all steaming modes.

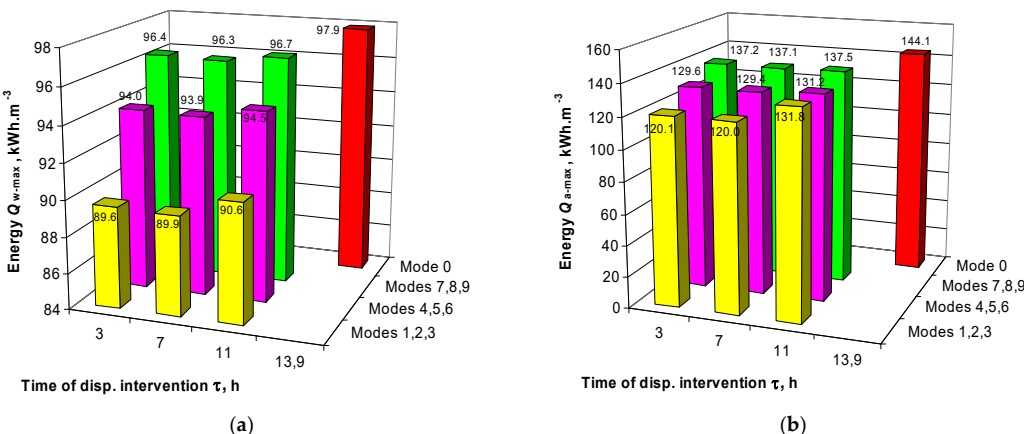

(**a**)                                                                (**b**)

**Figure 7.** Change in $Q_{\text{w-max}}$ (**a**) and $Q_{\text{a-max}}$ (**b**) of the studied steaming modes, depending on $t_{\text{m1}}$ and $\tau$.

In Figure 8 the change in $\eta$ of all studied modes is presented.

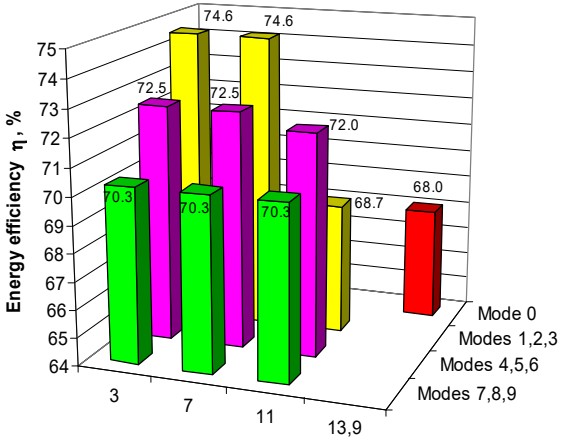

**Figure 8.** Change in $\eta$ of the studied modes, depending on $t_{\text{m1}}$ and $\tau$.

## 4. Discussion

It can be seen in Figure 3 that the temperature at the representative points of all the prisms changes along extremely complex curves, both during the steaming modes and during the subsequent conditioning of the heated prisms in an air environment. The temperature of the processing medium in the autoclave at the beginning of all modes rises gradually and reaches the maximum value of $t_{\text{m1}} = 130\ °C$ after 2.65 h.

The violation of the smoothness of the line $t_{\text{m}} = f(\tau)$ around the 1st h at the beginning of the modes is caused by the phenomenon of the uneven melting of the ice formed by the free water in the wood, which is adequately reflected in the mathematical models.

**Table 2.** Change in $\tau_{\text{steam}}$, $\tau_{\text{mode}}$, $t_{\text{w-avg}}$ at $\tau_2$, $Q_{\text{w-max}}$, $Q_{\text{a-max}}$, and $\eta$ of the studied steaming modes, depending on $\Delta\tau$.

| Steaming Modes | $\Delta\tau$, h | $\tau_2 = \tau_{\text{steam}}$, h | $\tau_4 = \tau_{\text{mode}}$, h | $t_{\text{w-avg}}$ at $\tau_2$, °C | $Q_{\text{w-max}}$, kWh·m$^{-3}$ | $Q_{\text{a-max}}$, kWh·m$^{-3}$ | $\eta$, % |
|---|---|---|---|---|---|---|---|
| Mode 0 | 0 | 13.9 | 17.4 | 91.7 | 97.95 | 144.08 | 68.0 |
| Mode 1 | 3 | 16.2 | 18.7 | 90.2 | 96.44 | 137.19 | 70.3 |
| Mode 2 | 7 | 15.7 | 18.2 | 90.0 | 96.34 | 137.06 | 70.3 |
| Mode 3 | 11 | 15.2 | 17.7 | 90.5 | 96.69 | 137.49 | 70.3 |
| Mode 4 | 3 | 18.7 | 20.2 | 87.3 | 94.01 | 129.65 | 72.5 |
| Mode 5 | 7 | 17.7 | 19.2 | 87.1 | 93.86 | 129.44 | 72.5 |
| Mode 6 | 11 | 16.7 | 18.2 | 87.9 | 94.50 | 131.21 | 72.0 |
| Mode 7 | 3 | 20.7 | 22.2 | 81.8 | 89.60 | 120.11 | 74.6 |
| Mode 8 | 7 | 19.1 | 20.6 | 81.7 | 89.53 | 119.98 | 74.6 |
| Mode 9 | 11 | 17.5 | 19.0 | 83.0 | 90.55 | 131.75 | 68.7 |

When this phenomenon occurs, a huge amount of heat is utilized to melt the aforementioned ice, which disturbs the thermal balance of the autoclave in the considered case of limited power of the steam generator, $q_{\text{source}}$, feeding it, and slows down the rise of $t_{\text{m}}$.

The duration of the basic Mode 0, in which there is no dispatcher intervention, is equal to 17.4 h, and the duration of the supply of steam to the autoclave in this mode is equal to 13.9 h. The earlier such an intervention is carried out and the greater the reduction in the maximum value of $t_{\text{m1}}$, the longer these durations are compared to those in the basic mode.

Figures 4–6 show that the increase in the steaming time causes a gradual smooth increase of $Q_{\text{w}}$ and $Q_{\text{a}}$ in the basic mode for the prisms' steaming. Due to the influence of the last 5 terms in the right-hand side of Equation (15), the rate of increase of $Q_{\text{a}}$ is greater than that of $Q_{\text{w}}$. This means that the differences between the corresponding graphs $Q_{\text{a}}$ and $Q_{\text{w}}$ are equal to the sum of the energies that provide the energy consumptions described by the last 5 terms in Equation (15).

The increase of $Q_{\text{w}}$ is analogous to that of $t_{\text{w-avg}}$ of the prisms in all investigated steaming modes, with and without dispatcher intervention.

The smoothness of increasing $Q_{\text{w}}$ and $Q_{\text{a}}$ during the steaming process is disturbed at the moments of application of dispatcher intervention in the steaming modes of the prisms. The earlier such intervention occurs or the greater the temperature change $\Delta t_{\text{m1}}$ in the mode, the more significant the difference between the values of $Q_{\text{w}}$ and $Q_{\text{a}}$ of the basic mode when compared to the corresponding mode with dispatcher intervention.

The thermal efficiency $\eta$ in the modes at changing operational conditions has values between 68.7% and 74.6%, while in the basic steaming mode, efficiency is equal to 68.0%. If the dispatcher's intervention was provided in the 3rd or 7th h of the modes, with a more significant reduction in $t_{\text{m1}}$, there was a greater increase in $\eta$ compared to $\eta$ of the basic mode.

When applying a dispatcher interference closer to the end of the modes, for example at the 11th h of the modes, this dependence is broken in the mode, with the smallest investigated decrease in $t_{\text{m1}}$ being from 130 to 120 °C.

When the efficiency $\eta$ is equal to 0.67, i.e., it is quite a bit greater than that of the basic steaming mode. The reason for this is the much higher $Q_{\text{a-max}}$ value at the 11th h compared to the $Q_{\text{a-max}}$ values at the 3rd and 7th h of the modes with a decrease in $t_{\text{m1}}$ from 130 to 120 °C.

## 5. Conclusions

This paper considers a methodology for computing the energy consumption and thermal efficiency of autoclaves during the treatment with saturated water vapor of frozen prisms in veneer production at changing operational conditions. When such conditions occur, the parameters of the autoclave's steaming modes have to be changed by the dispatcher

or control system in such a way as to ensure optimal plasticity of the wood immediately before cutting the veneer.

The change in the maximum values of $Q_w$ and $Q_a$, respectively $Q_{w\text{-max}}$ and $Q_{a\text{-max}}$, and with their change, also that of of $\eta$, calculated during computer simulations with two own coupled models for each of the investigated modes for autoclave steaming of beech prisms with industrial parameters, led to the following conclusions:

- At the moment $\tau_2 = \tau_{steam} = 13.9$ h, when the introduction of water vapor into the autoclave ends, the greatest values $Q_{w\text{-max}} = 97.95$ kWh·m$^{-3}$ and $Q_{a\text{-max}} = 144.08$ kWh·m$^{-3}$ are established in the basic mode, which takes place at $t_{m1} = 130\,°C = $ const. These values of $Q_{w\text{-max}}$ and $Q_{a\text{-max}}$ determine the presence of the lowest value of $\eta = 68.0\%$ of the basic mode compared to the thermal efficiency of all modes with dispatcher intervention.

- When, upon application of dispatcher intervention, the temperature of the processing medium in the autoclave is reduced from $t_{m1} = 130\,°C$ to $t_{m1} = 120\,°C$, the energies $Q_w$ and $Q_a$ reach their maximum values at moments $\tau_2 = \tau_{steam}$, which depend on the occurrence times of this intervention. Then, they are equal to about $Q_{w\text{-max}} \approx 96.5$ kWh·m$^{-3}$ and $Q_{a\text{-max}} \approx 137.3$ kWh·m$^{-3}$, respectively. As a result, the energy efficiency turns out to be the same, equal to 70.3% for all three such modes investigated.

- When, after dispatcher intervention, the temperature $t_{m1}$ is reduced from 130 to 110 °C, the energies $Q_w$ and $Q_a$ reach maximum values at the moment $\tau_2 = \tau_{steam}$ only at $\Delta\tau_a = 3$ h and $\Delta\tau_b = 7$ h. Then, they are equal to about $Q_{w\text{-max}} \approx 93.9$ kWh·m$^{-3}$ and $Q_{a\text{-max}} \approx 129.5$ kWh·m$^{-3}$, respectively, resulting in $\eta \approx 72.5\%$. In the case when $\Delta\tau_c = 11$ h, the maximum values of $Q_{w\text{-max}} = 94.6$ kWh·m$^{-3}$ and $Q_{a\text{-max}} = 131.2$ kWh·m$^{-3}$ are reached at the moment of application of the dispatcher intervention, and this causes a reduction of $\eta$ to $\eta = 72.0\%$. In this case $\tau_{steam} = 16.7$ h.

- When, after dispatcher intervention, $t_{m1}$ is reduced from 130 to 100 °C, $Q_w$ and $Q_a$ reach maximum values at the moment $\tau_2 = \tau_{steam}$ also only at $\Delta\tau_a = 3$ h and $\Delta\tau_b = 7$ h. Then, they are equal to approximately $Q_{w\text{-max}} \approx 89.6$ kWh·m$^{-3}$ and $Q_{a\text{-max}} \approx 120.1$ kWh·m$^{-3}$, respectively, resulting in $\eta \approx 74.6\%$. In the case when $\Delta\tau_c = 11$ h, the maximum values of $Q_{w\text{-max}} = 90.6$ kWh·m$^{-3}$ and $Q_{a\text{-max}} = 131.8$ kWh·m$^{-3}$ are reached at the time of application of the dispatcher intervention and this causes a reduction of $\eta$ to $\eta = 68.7\%$.

The presented methodology can be applied in the creation of system software for model-based energy-efficient automatic control of technologies for the water vapor treatment of frozen and non-frozen wood materials with a desired duration of modes, set by a dispatcher.

**Author Contributions:** Conceptualization, N.D., P.N. and D.A.; methodology, N.D., P.N. and D.A.; software, N.D.; validation, N.D., P.V. and N.T.; formal analysis, P.N. and N.D.; investigation, N.D., P.N., P.V. and D.A.; resources, N.D., P.N. and D.A.; data curation, N.D. and N.T.; writing—original draft preparation, N.D. and D.A.; writing—review and editing, N.D., P.N. and D.A.; visualization, N.T., P.V. and N.D.; supervision, N.D.; project administration, N.D.; funding acquisition, N.D. All authors have read and agreed to the published version of the manuscript.

**Funding:** This research received no external funding.

**Institutional Review Board Statement:** Not applicable.

**Informed Consent Statement:** Not applicable.

**Conflicts of Interest:** The authors declare no conflict of interest.

## Abbreviations

**Symbols**

| | |
|---|---|
| $b$ | width of the wooden prisms, m |
| $c$ | specific heat capacity, $J \cdot kg^{-1} \cdot K^{-1}$ |
| $d$ | thickness of the prisms, m |
| $D$ | diameter of the steaming autoclave, m |
| $l$ | length of the prisms, m |
| $L$ | length of the autoclave, m |
| $q$ | thermal power, kW |
| $Q$ | thermal energy, $kWh \cdot m^{-3}$ |
| $S$ | aria, $m^2$ |
| $T$ | temperature, K |
| $t$ | temperature, °C: $t = T - 273.15$ |
| $u$ | moisture content, $kg \cdot kg^{-1}$ = %/100 |
| $x$ | coordinate along $d$ |
| $y$ | coordinate along $b$ |
| $\alpha$ | convective heat transfer coefficient, $W \cdot m^{-2} \cdot K^{-1}$ |
| $\gamma$ | loading of the autoclave, $m^3 \cdot m^{-3}$ = %/100 |
| $\eta$ | energy efficiency, % |
| $\lambda$ | thermal conductivity, $W \cdot m^{-1} \cdot K^{-1}$ |
| $\rho$ | density, $kg \cdot m^{-3}$ |
| $\tau$ | time, s |
| $\Delta\tau$ | step along $\tau$, s |

**Subscripts:**

| | |
|---|---|
| a | autoclave |
| ad | anatomical direction (for wood) |
| avg | average |
| b | basic (for density or for steaming mode) |
| bw | bound water |
| cr | cross sectional to the fibers |
| cw | condensed water (for autoclave) |
| e | emission (for autoclave) |
| eff1 | effective (for $c$ of wood with frozen bound water) |
| eff2 | effective (for $c$ of wood with frozen free water) |
| eff3 | effective (for $c$ of non-frozen wood) |
| fr | frozen |
| fv | free volume (for autoclave) |
| fw | free water |
| ice | ice |
| il | insulating layer |
| h | heat |
| $i$ | mesh point along $x$ |
| $j$ | mesh point along $y$ |
| m | medium |
| mb | metal body (for autoclave and trolleys in it for placing of wood materials) |
| nfr | non-frozen |
| s | surface |
| w | wood |
| 0 | initial |

**Superscripts:**

| | |
|---|---|
| $n$ | time level: $n = 0, 1, 2, 3, \ldots, \tau_{end}/\Delta\tau$ |
| 272.15 | at 272.15 K, i.e., at $-1$ °C |
| 293.15 | at 293.15 K, i.e., at 20 °C |

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
