# Peer review of "Computing the Thermal Efficiency of Autoclaves during Steaming of Frozen Prisms for Veneer Production at Changing Operational Conditions"

_processes, doi:10.3390/pr11030822_

Round 1

Reviewer 1 Report

The manuscript titled Computing the Thermal Efficiency of Autoclaves during Steaming of Frozen Prisms for Veneer Production at Changing Operational Conditions. The ideas in the manuscript are very interesting. This manuscript is well structured and well written, which is easy to follow. The figures and tables are neat and easy to understand. The methodology is thoroughly explained and the work overall seems to be skillfully performed. 

Generally, I suggest this manuscript can be accepted for publication after minor revision.

It is recommended to take notice the following:

(1) The “introduction” focuses on telling the readers research status of the subject, existing problem, why you do this work or the importance of your work, and what problem you will solve (Objective). Thus, modify the introduction

(2) Please tell the readers why you select the levels of parameters, from the published papers, industrial machining or others? The papers you published should help every reader better understand what you written.

(3) Provide more information for the "Change in η of the studied modes, depending on tm1 and τ"

(4) It is hard to follow the "Figures 3 - 5 show that the increase in the steaming time causes a gradual smooth increase of Qw and Qa in the basic mode for prisms’ steaming", make more explanation.

(5) On Figure 8→In Figure 8

(6) Supplement the limitation of your work.

(7) Conclusion should be succinct.

(8) Reference format should be modified based on MDPI journal requirement.

Author Response

Dear Referee 1,

We highly appreciate your efforts to read carefully and to assess our manuscript "Computing the Thermal Efficiency of Autoclaves during Steaming of Frozen Prisms for Veneer Production at Changing Operational Conditions", and also to help us with its improvement. Many thanks for your comments and recommendations.

The revised manuscript contains changes and some new text, which are given in Red color.

The following amendments were made to the manuscript according to your recommendations:

  1. A sentence was added to the Introduction that reinforces the arguments and the need to conduct the research described in the manuscript.
  2. In subsections 2.1 and 2.4, a total of 4 new sentences have been added explaining the reasons for choosing the specified parameters of the research objects.
  3. In the Discussion section, 5 new sentences have been added, clarifying the variation of the thermal efficiency η of the studied modes depending on tm1 and τ.
  4. On line 196 of the manuscript the text "In Figures 3-5" is replaced by the correct "In Figures 4-6".
  5. The Conclusions section has been reduced by a total of 3 sentences.
  6. The requirement that the main text of the manuscript be at least 4000 words is satisfied as follows: the added sentences contain a total of 321 words and the Conclusions section is reduced by a total of 75 words. As a result, the revised version of the manuscript contained 4135 words.
  7. All references are written according to the "Processes template 2023" available on the Internet.

Reviewer 2 Report

No Suggestions

Author Response

Dear Referee 2,

We highly appreciate your efforts to read carefully and to assess our manuscript "Computing the Thermal Efficiency of Autoclaves during Steaming of Frozen Prisms for Veneer Production at Changing Operational Conditions", and also to help us with its improvement. Many thanks for your comments and recommendations.

The revised manuscript contains changes and some new text according to the recommendations of all 4 reviewers, which are given in Red color.

Reviewer 3 Report

The authors must complete with 10 bibliographic references from the field of research, the last 5 years.

Author Response

Dear Referee 3,

We highly appreciate your efforts to read carefully and to assess our manuscript "Computing the Thermal Efficiency of Autoclaves during Steaming of Frozen Prisms for Veneer Production at Changing Operational Conditions", and also to help us with its improvement. Many thanks for your comments and recommendations.

The revised manuscript contains changes and some new text according to the recommendations of all 4 reviewers, which are given in Red color.

Based on your recommendation, the text of the manuscript was re-spell-checked and the spelling of 6 words in it was improved.

In our opinion, your recommendation to limit the number of references in the manuscript to 10, and that published only in the last 5 years, would strongly limit the described background and field of the research. Since none of the other 3 reviewers made such a suggestion, the list of references was left unchanged.

Reviewer 4 Report

The manuscript is dealing with a topic of practical relevance, using a good scientific background.

After the following suggested minor corrections are made, I suggest to accept the manuscript for publication.

Line 11: Please delete the duplication of "Correspondence"

Subscript "fr" is missing from the list of subscripts.

Table 1: First column is duplicated (Steaming modes)

Figures 7-8: However, you show the parameters on the horizontal axes in the figure that indcates the steaming modes given in Table 1, it would help the understanding if you add the steaming mode in legend. For example by using different color for each column of the diagram.

Author Response

Dear Referee 4,

We highly appreciate your efforts to read carefully and to assess our manuscript "Computing the Thermal Efficiency of Autoclaves during Steaming of Frozen Prisms for Veneer Production at Changing Operational Conditions", and also to help us with its improvement. Many thanks for your comments and recommendations.

The revised manuscript contains changes and some new text according to the recommendations of all 4 reviewers, which are given in Red color.

The following amendments were made to the manuscript according to your recommendations:

  1. On line 11 of the manuscript, the duplication of "Correspondence" has been removed.
  2. The "fr" tag was added to the Subscripts list.
  3. The duplication of the first column in Table 1 was removed.
  4. In the legends of Figures 7-9, the names of the researched steaming modes were added